# Determining the timing of pubertal onset via a multicohort analysis of growth

Essi Syrjälä[1]*, Harri Niinikoski[2,3,4,5°], Helena E. Virtanen[2°], Jorma Ilonen[6,7], Mikael Knip[8,9,10,11], Nina Hutri-Kähönen[12], Katja Pahkala[4,5,13], Olli T. Raitakari[4,5,14], Wiwat Rodprasert[2], Jorma Toppari[2,3,5], Suvi M. Virtanen[1,11,15,16], Riitta Veijola[17,18], Jaakko Peltonen[19], Jaakko Nevalainen[1]

1 Faculty of Social Sciences, Unit of Health Sciences, Tampere University, Tampere, Finland, 2 Research Centre for Integrative Physiology and Pharmacology, Institute of Biomedicine, University of Turku, Turku, Finland, 3 Department of Pediatrics, Turku University Hospital, Turku, Finland, 4 Research Centre of Applied and Preventive Cardiovascular Medicine, University of Turku, Turku, Finland, 5 Centre for Population Health Research, University of Turku and Turku University Hospital, Turku, Finland, 6 Immunogenetics Laboratory, Institute of Biomedicine, University of Turku, Turku, Finland, 7 Clinical Microbiology, Turku University Hospital, Turku, Finland, 8 Pediatric Research Center, Children's Hospital, University of Helsinki and Helsinki University Hospital, Helsinki, Finland, 9 Research Program for Clinical and Molecular Metabolism, Faculty of Medicine, University of Helsinki, Helsinki, Finland, 10 Folkhälsan Research Center, Helsinki, Finland, 11 Center for Child Health Research, Tampere University and Tampere University Hospital, Tampere, Finland, 12 Department of Pediatrics, Tampere University and Tampere University Hospital, Tampere, Finland, 13 Paavo Nurmi Centre, Sports and Exercise Medicine Unit, Department of Physical Activity and Health, University of Turku, Turku, Finland, 14 Department of Clinical Physiology and Nuclear Medicine, Turku University Hospital, Turku, Finland, 15 Tays Research, Development and Innovation Center, Tampere University Hospital, Tampere, Finland, 16 Health and Well-Being Promotion Unit, Department of Public Health and Welfare, Finnish Institute for Health and Welfare, Helsinki, Finland, 17 Department of Pediatrics, PEDEGO Research Unit, Medical Research Center, University of Oulu, Oulu, Finland, 18 Department of Children and Adolescents, Oulu University Hospital, Oulu, Finland, 19 Faculty of Information Technology and Communication Sciences, Tampere University, Tampere, Finland

☯ These authors contributed equally to this work.
* essi.syrjala@tuni.fi

**Data Availability Statement:** The dataset supporting the conclusions of this article were obtained from the DIPP, STRIP and Boy cohort studies. The dataset comprises health related

## Abstract

### Objective

Growth-based determination of pubertal onset timing would be cheap and practical. We aimed to determine this timing based on pubertal growth markers. Secondary aims were to estimate the differences in growth between cohorts and identify the role of overweight in onset timing.

### Design

This multicohort study includes data from three Finnish cohorts—the Type 1 Diabetes Prediction and Prevention (DIPP, $N = 2,825$) Study, the Special Turku Coronary Risk Factor Intervention Project (STRIP, $N = 711$), and the Boy cohort ($N = 66$). Children were monitored for growth and Tanner staging (except in DIPP).

### Methods

The growth data were analyzed using a Super-Imposition by Translation And Rotation growth curve model, and pubertal onset analyses were run using a time-to-pubertal onset model.

participant data and their use is therefore restricted under the regulations on professional secrecy (Act on the Openness of Government Activities, 612/1999) and on sensitive personal data (Personal Data Act, 523/1999, implementing the EU data protection directive 95/46/EC). Due to these legal restrictions, the data from this study cannot be stored in public repositories or otherwise made publicly available. However, data access may be permitted on a case by case basis upon request only. Data sharing outside the research groups is done in collaboration with DIPP, STRIP and Boy cohort groups and requires a group-specific data transfer agreements (DTA). Investigators can submit an expression of interest to the chairman of the DIPP steering committee (Prof Riitta Veijola, University of Oulu, Oulu, Finland. Email: riitta.veijola@oulu.fi), to the chairman of the STRIP steering group (Prof Olli Raitakari, University of Turku, Turku, Finland. Email: olli.raitakari@utu.fi) and to the chairman of the Boy cohort steering group (Prof Jorma Toppari, University of Turku, Turku, Finland. Email: jorma.toppari@utu.fi).

**Funding:** The DIPP Study was supported by the Academy of Finland (grants 63672, 68292, 79685, 79686, 80846, 114666, 126813, 129492, 139391, 201988, 210632, 276475, 308066, and Center of Excellence in Molecular Systems Immunology and Physiology Research 2012-2017, Decision No. 250114), URL: https://www.aka.fi/en/; European Foundation for the Study of Diabetes (EFSD) award supported by EFSD/JDRF/Lilly, URL: http://www.europeandiabetesfoundation.org/; the Finnish Diabetes Association, URL: https://www.diabetes.fi/en/finnish_diabetes_association; the Finnish Diabetes Research Foundation, URL: https://www.diabetestutkimus.fi/en; the Juho Vainio Foundation, URL: https://juhovainionsaatio.fi/en/juho-vainio-foundation/; the Juvenile Diabetes Research Foundation International (grants 4-1998-274, 4-1999-731, and 4-2001-435), URL: https://www.jdrf.org/; the Competitive State Research Financing of the Expert Responsibility area of Tampere University Hospital (grants 9E082, 9F089, 9G087, 9H092, 9J147, 9K149, 9L042, 9L117, 9M114, 9N086, 9P057, 9R055, 9S074, 9U065, and 9V072); Oulu University Hospital Research Funds; Turku University Hospital (state research funding ERVA); the European Comission (grant BMH4-CT98-3314), URL: https://ec.europa.eu/info/index_en; the Novo Nordisk Foundation, URL: https://novonordiskfonden.dk/en/; Special Research Funds for University Hospitals in Finland; and the Sigrid Jusélius Foundation, URL: https://www.sigridjuselius.fi/en/. The STRIP was supported by the Academy of Finland (grants 251360, 276861, 307996, and 322112), URL: https://www.aka.fi/en/;

## Results

The time-to-pubertal onset model used age at peak height velocity (aPHV), peak height velocity (PHV), and overweight status as covariates, with interaction between aPHV and overweight status for girls, and succeeded in determining the onset timing. Cross-validation showed a good agreement (71.0% for girls, 77.0% for boys) between the observed and predicted onset timings. Children in STRIP were taller overall (girls: 1.7 [95% CI: 0.9, 2.5] cm, boys: 1.0 [0.3, 2.2] cm) and had higher PHV values (girls: 0.13 [0.02, 0.25] cm/year, boys: 0.35 [0.21, 0.49] cm/year) than those in DIPP. Boys in the Boy cohort were taller (2.3 [0.3, 4.2] cm) compared with DIPP. Overweight girls showed pubertal onset at 1.0 [0.7, 1.4] year earlier compared with other girls. In boys, there was no such difference.

## Conclusions

The novel modeling approach provides an opportunity to evaluate the Tanner breast/genital stage–based pubertal onset timing in cohort studies including longitudinal data on growth but lacking pubertal follow-up.

## Introduction

Puberty is a period of rapid growth involving changes in hormonal activity. Most recent studies on pubertal timing have reported a shift of puberty toward earlier ages during the last decades, especially in girls [1,2]. The commonly used marker of pubertal onset in girls is the development of breasts [3]. The emergence of breast buds defines Tanner breast stage 2 (Tanner B2) [4]. In boys, the commonly used marker of pubertal onset is testicular enlargement [3]. An enlargement to the volume of $> 3$ mL defines Tanner genital stage 2 (Tanner G2) [4], and this volume is consistent with a testis length $\geq 25$ mm [5]. However, puberty-induced body changes are a sensitive issue for children and adolescents, and thus, the physical measurement of pubertal onset may be challenging; at the same time, indirect measurement—for example, by hormone levels—is expensive.

One well-known feature of puberty is the adolescent growth spurt. The timing and intensity of the peak of the growth spurt are measured by the peak height velocity (PHV, cm/year) and age at PHV (aPHV, years). Estimation of the PHV and aPHV requires the collection of repeated height measurements and an appropriate analysis methodology. Multiple mathematical models have been proposed to describe adolescent growth [6–10]; the Super-Imposition by Translation And Rotation (SITAR) growth curve model is one of the most recent [11]. The SITAR model has been shown to efficiently summarize growth in adolescence [8] and determine the PHV and aPHV. In addition, it has been suggested that joint modeling of several cohorts with the SITAR model can improve the fit, and that the model can straightforwardly produce cohort-specific growth parameters [12].

In some studies, the start of the growth spurt has been used as the first sign of pubertal maturation [9,10]. In addition, different markers for different stages of pubertal development have been used in the existing literature [13,14], and growth in association with pubertal development has been widely studied [15–19]. However, the literature does not present the relation of the precisely estimated aPHV and PHV with the age at pubertal onset as a continuous variable, with consideration of interval censoring. When researchers are interested in determining the age at pubertal onset as defined by the Tanner breast/genital stages, but the stages have not

the Juho Vainio Foundation, URL: https://
juhovainionsaatio.fi/en/juho-vainio-foundation/; the
Finnish Foundation for Cardiovascular Research,
URL: https://www.sydantutkimussaatio.fi/en/
foundation; the Finnish Ministry of Education and
Culture, URL: https://minedu.fi/en/frontpage; the
Finnish Cultural Foundation, URL: https://skr.fi/en;
the Sigrid Jusélius Foundation, URL: https://www.
sigridjuselius.fi/en/; Special Governmental Grants
for Health Sciences Research (Turku University
Hospital); the Yrjö Jahnsson Foundation, URL:
https://www.yjs.fi/en/; and the Turku University
Foundation. The Boy cohort study was supported
by Turku University Hospital (state research
funding ERVA); the Sigrid Jusélius Foundation,
URL: https://www.sigridjuselius.fi/en/; the Novo
Nordisk Foundation (grant NNF16OC0021302),
URL: https://novonordiskfonden.dk/en/; European
Commission (grants BMH4-CT96-0314, QLK4-CT-
1999-01422, QLK4-CT-2001-00269, QLK4-2002-
00603, and FP7/2008-2012: DEER 212844), URL:
https://ec.europa.eu/info/index_en; the Academy of
Finland (grants 77320, 211480, 121880, 136850,
253341, 308065, and 128576), URL: https://www.
aka.fi/en/; and the Foundation for Pediatric
Research, URL: https://www.
lastentautientutkimussaatio.fi/. The funders had no
role in study design, data collection and analysis,
decision to publish, or preparation of the
manuscript.

**Competing interests:** The authors have declared
that no competing interests exist.

been measured, determination based on pubertal growth markers becomes an attractive possibility. Since height data are often recorded as a part of routine health surveillance and collected in many clinical follow-up studies, it is generally possible to determine the aPHV and PHV. aPHV is a well-known milestone of puberty, and it correlates with the age at pubertal onset, as does PHV [20,21].

Previous studies have indicated differences in anthropometrics between regions in Finland. For example, adults living in Southwest Finland have been found to be taller than adults living in Northern Ostrobothnia [22]. The present study can shed light on whether this difference can be seen during adolescence.

The association between an increased body mass index (BMI) in childhood and a notable shift to earlier puberty has been widely reported for girls [23,24]. For boys, the evidence of such an association is contradictory [23], with some studies suggesting the association to earlier [24,25] and some studies to later pubertal onset [26]. There is also evidence for different directions of association depending on the magnitude of increase in BMI [27].

The primary aim of the present multicohort study was to estimate the pubertal growth markers, aPHV and PHV, and to determine the timing of pubertal onset based on them. The secondary aims were to investigate whether there are differences in growth during adolescence between three recent Finnish child and adolescent studies and to assess the role of being overweight or obese in the timing of pubertal onset.

## Material and methods

### Study population and design

The data of the current multicohort study are derived from three Finnish cohort studies—the Type 1 Diabetes Prediction and Prevention (DIPP) Study [28], the Special Turku Coronary Risk Factor Intervention Project (STRIP) [29,30], and the Boy cohort [31]. Multi-cohort study data were collected in two different regions and during different time periods (Table 1).

The DIPP Study is a Finnish prospective population-based birth cohort study. Newborn infants born in Oulu, Tampere and Turku University Hospitals were screened for *HLA-DQB1*-conferred susceptibility to type 1 diabetes using cord blood samples. Infants carrying increased genetic susceptibility were recruited for regular follow-up. The current study includes 2825 children born in Oulu University Hospital.

The STRIP is a Finnish prospective randomized intervention trial aimed at preventing coronary heart disease and atherosclerosis by a lifestyle-based intervention. One thousand sixty-two 7-month-old infants born in 1989–1991 were recruited at well-baby clinics in Turku. They were randomized into a life-style based intervention group (n = 540) or a control group (n = 522).

The Boy cohort is a Finnish prospective population-based birth cohort study on the prevalence of congenital cryptorchidism. It included 1494 antenatally recruited infants born 1997–1999 in Turku University Hospital. In addition, 184 boys born in 1997–2002 were included in a case-control study on risk factors of congenital cryptorchidism. Of all the boys, 119 participated in a pubertal follow-up with 52 of them having (cases) and 67 not having (controls) a history of congenital cryptorchidism. The data of the controls in the pubertal follow-up study were used in the current analysis.

Children in all cohorts were monitored for growth; children in the STRIP and Boy cohort were also monitored for pubertal markers (Tanner staging) from the age of 8 years. Tanner staging included the following assessments:

**Table 1. Characteristics of the cohort studies.**

|  | DIPP | STRIP | Boy cohort |
|---|---|---|---|
| **City** | Oulu | Turku | Turku |
| **N, growth data** | 2825 | 711 | 66 |
| **N, pubertal data** | - | 706 | 66 |
| **Recruitment** | 1995–2011 | 1990–1992 | 1997–2002 |
| **Follow-up frequency for growth** | Every 6–12 mo | Every 6–12 mo | Every 6 mo |
| **Follow-up frequency for pubertal markers** | - | Every 12 mo | Every 6 mo |
| **Recruited children** | Newborn infants with HLA-DQB1-conferred susceptibility to type 1 diabetes | 7-month-old infants | Newborn male infants |

- Breast development in girls, evaluated by palpation and measurement of breast tissue with a ruler (Tanner B).

- External genitalia in boys, determined by measuring the length of the testes with a ruler (Tanner G), as described in Sadov et al. [31]

- Pubic hair for both sexes (Tanner pubic hair stage).

The DIPP Study was included in the analyses based on the further interest to determine the timing of pubertal onset based on growth data in the DIPP children and to improve the fit of the growth curves [12].

## Ethics

The DIPP Study (Oulu) was approved by the Ethical Committee of the Faculty of Medicine, University of Oulu, and the Ethical Committee of the city of Oulu. Parents gave their written informed consent first for the genetic testing of the newborn infant and then again for participation in the follow-up.

The STRIP was approved by the Joint Committee on Ethics of the Turku University and the Turku University Central Hospital. Parents of the children gave their written informed consent.

The original birth cohort study, the case-control study and the pubertal follow-up study of the Boy cohort were approved by the Joint Committee on Ethics of the Turku University and the Turku University Central Hospital. Parents of the children gave their written informed consent and the participants of the pubertal follow-up gave their assent for the study.

All the cohorts adhere to the Declaration of Helsinki. Data were pseudonymized before the corresponding author accessed them for the analyses. The written informed consents given within the cohort studies enable the usage of collected data in research, including the current multi-cohort study.

## Defining the age at pubertal onset

For girls, pubertal onset was determined based on Tanner breast stages. The onset of puberty was considered to occur between the last observed Tanner B1 and the first observed Tanner B2. In a time-to-pubertal onset model, the transition between the stages was understood as an interval censored between those points of time. If a girl had missing Tanner B2 measurements, but Tanner B3, B4, and/or B5 measurements were available, the time interval between the last Tanner B1 and the next available Tanner B measurement was used.

In the original data provided for boys, Tanner G2 was considered to occur when testicular length was $\geq 20$ mm. However, instead of using this old threshold, based on the current recommendation of a testis length of 25 mm as the marker of pubertal onset,[5] ages at pubertal onset were recalculated using testis length measurements. The pubertal onset was considered to occur when the length of the larger testis was $\geq 25$ mm at two successive visits [31]. Then, the age at pubertal onset was considered as the interval censored between the last $< 25$ mm and the first (of the two successive visits) $\geq 25$ mm measurement in the time-to-pubertal onset model [32].

If the last measurement was taken before pubertal onset, the age at pubertal onset was handled as right censored: The child had dropped out before pubertal onset, and we only observed that the onset had occurred after that age [32]. If there were no measurements before pubertal onset had taken place, the age at pubertal onset was handled as left censored: We only observed that the age at pubertal onset was less than the age at first measurement [32].

## Statistical analyses

The analyses were performed in two stages and conducted separately for boys and girls with R 3.5.3 [33] using the "sitar" function from the "sitar" package [34], the "survreg" function from the "survival" package [35], and the "glmnet" function from the "glmnet" package [36].

Data from all the cohort studies were used in the first stage for SITAR growth curve analyses to investigate the differences in growth between the cohort studies. All growth data from individuals between the ages of 7 and 19 were included. In the second stage, for the time-to-pubertal onset analyses, only data from the STRIP and Boy cohort were used because they included measurements on pubertal development. In this analysis, children with at least four height measurements were included. This decision was based on the visual inspection of the distributions of the aPHV values by different numbers of height measurements (S1 Fig). The variation in aPHV values was minimal with smaller number of height measurements indicating that the aPHV estimation is based more on the average growth curve than on an individual curve.

**First stage: Super-Imposition by Translation And Rotation (SITAR) growth curve model.** In the first stage, growth curves were estimated with the SITAR growth curve model, which has been shown to efficiently summarize growth in adolescence using the three dimensions of size, timing, and intensity [8,11]. Size corresponds to the mean height, timing to the aPHV, and intensity to the PHV. The model estimates a mean growth curve through fixed-effect parameters for the dimensions. It also estimates the random-effect parameters, representing the shift from the mean curve for each individual. The random effect for size represents a shift up or down of the height curve of a particular child relative to overall mean height curve (measured in centimeters). The random effect for timing represents a shift earlier or later of the aPHV of a particular child relative to mean APHV (measured in years). The random effect for intensity represents a change of the PHV of a particular child relative to mean PHV (can be viewed as a percentage difference when multiplied by 100). Individual growth curves are estimated through the fixed-effect and random-effect parameters.

The mean aPHV and PHV were derived by differentiating the mean height curve estimated by the SITAR model and by identifying the age at which the derivative curve reached its maximum. The PHV was the value of the derivative at that age. Individual aPHVs and PHVs were derived from the individual height curves in the same way. The differences in growth between cohorts were investigated based on the cohort-specific fixed-effect parameters of the SITAR model. Full technical details of the SITAR modeling are provided in S1 File.

**Overweight classification.** The overweight status of each child was used as a predictor in the time-to-pubertal onset models to investigate the role of being overweight in the timing of pubertal onset. The overweight status was determined based on the BMI when the children's growth spurt started. The start of the growth spurt was defined as the lowest point in the derivative of the individual growth curve (velocity curve). The corresponding BMI for the child was obtained from the individual BMI–age trajectory fitted by a polynomial spline mixed-effects model. The BMI–age mixed-effects model included the random intercept and random effects for all the polynomial regression coefficients. Children were classified into two classes based on the obtained BMI value as follows: overweight (ISO-BMI > 25, including overweight and obese) and other (including normal weight and underweight) children, based on the Finnish ISO-BMI classification for children [37].

**Second stage: Time-to-pubertal onset model.** In the second stage, two time-to-pubertal onset models—an extended model and a simple model—were fitted to determine the timing of pubertal onset based on growth markers and to study the role of overweight in the timing. In both models, the outcome was the timing of pubertal onset. The simple model used only aPHV and PHV as predictors. The purpose of the model was to study whether aPHV and PHV alone could determine the timing of pubertal onset. The extended model included all the important variables, which were selected using a lasso regression technique with a time-to-pubertal onset model. The considered variables were the aPHV, PHV, and size parameter ($a_i$) obtained from the SITAR model; the overweight status of the child; and all the second-order interactions between the variables. Full technical details of the time-to-pubertal onset models are provided in S1 File.

*Determination of the predictive ability of the model.* The predictive ability of the time-to-pubertal onset model was determined using 10-fold cross-validation to obtain the predicted pubertal onsets. Using cross-validation, we could investigate the predictive ability in external data more reliably because the prediction was always done for children who were not included in the estimation of the model. Ten-fold cross-validation was chosen because it has been indicated to be one of the methods with the smallest bias [38].

The estimated individual means at given values of the predictors were used as predicted pubertal onsets. Prediction success was evaluated by comparing the observed pubertal onset age interval to a year length interval centered on the predicted age at pubertal onset. A year length interval was chosen to agree with the prescheduled visits for most of the children. If those intervals overlapped, it was considered to represent an agreement. Only children with an observed interval length of less than 1.5 years were included in the evaluation of success. To visualize the success, prediction intervals were obtained using the appropriate quantiles of the normal distribution centered on the individual prediction and with an estimate of the variance ($\hat{\sigma}^2$).

## Results

### Determination of the timing of pubertal onset based on growth markers

The mean age at pubertal onset was 10.5 years (SD: 1.25) for STRIP girls and 11.7 (SD: 0.90) and 11.3 (SD: 0.98) years for STRIP and Boy cohort boys, respectively, as estimated from separate cohort-specific time-to-pubertal onset models. Of the 340 girls and 432 boys with pubertal data, 306 girls (90.0%) and 392 boys (90.7%) fulfilled the criterion of four height measurements and were included in the time-to-pubertal onset analyses (Stage 2). Of those children, for 7 (2.3%) girls and 40 (10.2%) boys, pubertal onset occurred after the last follow-up visit (right censored), and for 4 (1.3%) girls, it occurred before the first follow-up visit (left censored).

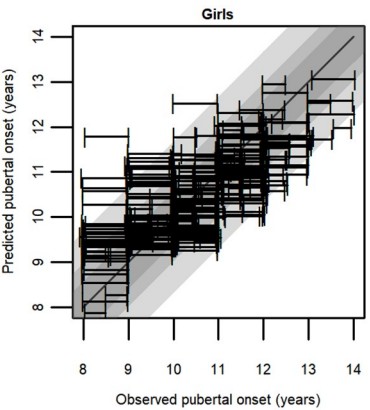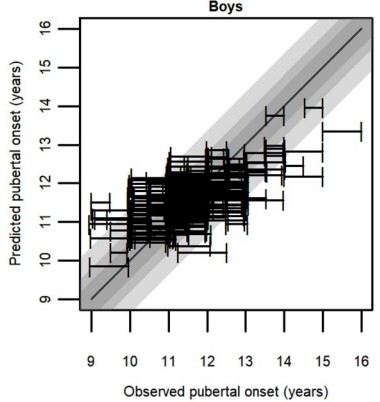

**Fig 1. Observed pubertal onset intervals plotted against the predicted pubertal onsets for girls and boys.** Grey areas represent 50%, 75% and 95% prediction intervals.

aPHV had an apparent direct association and PHV an apparent inverse association with age at pubertal onset in both sexes (S2 Fig). Only the size parameter was not associated with the timing of pubertal onset (S2 Fig). For both sexes, the important variables chosen for the extended time-to-pubertal onset model were aPHV, PHV, and overweight status. For girls, the interaction between aPHV and overweight status was added to the model because the association of aPHV to the timing of pubertal onset appeared to differ depending on the overweight status (S3 Fig). aPHV clearly dominated the prediction over PHV for both sexes. All the time-to-pubertal onset model parameters and prediction results are presented in S1 Table. The predictive ability of the extended model was only slightly better than the predictive ability of the simple model for both sexes (S1 Table).

The extended model predictions tended to fall in the center of the data, suggesting good predictive ability for the test data (Fig 1). Most of the observed ages at pubertal onset overlapped with the prediction intervals (Fig 1), indicating that the predicted onset falls close to the observed onset. The agreement for girls was 71.0%, and for boys, it was 77.0%.

## Differences in growth between cohorts

Of the children, 1,671 girls and 2,162 boys had growth data and were included in the growth analyses (Stage 1). The median numbers of height measurements were 7 (interquartile range, IQR: 3–9) and 11 (IQR: 10–20) for girls in DIPP and STRIP, respectively, while they were 7 (IQR: 3–9), 11 (IQR: 9–19), and 16 (IQR: 14–18) for boys in the DIPP, STRIP, and Boy cohort, respectively.

Both girls and boys in the STRIP were taller and had higher PHV values compared with those in DIPP (Table 2). In addition, boys in the Boy cohort were taller compared with those in DIPP (Table 2). There were no differences in aPHV values between the cohorts (Table 2). The boys in the STRIP and the Boy cohort came from the same region in Southwest Finland, whereas the DIPP cohort presented boys from Northern Finland. Cohort-specific growth and velocity curves illustrate the differences between cohorts (Fig 2).

## The role of being overweight in the timing of pubertal onset

Of the girls, 51 (15.0%) were overweight. Girls with overweight reached puberty earlier than other girls did (−1.0 [95% CI: −1.4, −0.7] years). However, the improvement in overall predictive ability was modest: The agreement increased from 68.1% to 71.0% while adding

**Table 2. Summary statistics of the growth markers by cohorts and by sexes; aPHV, PHV, estimated height at 8 years and estimated height at 18 years.**

| | Girls | | | | Boys | | | | | | | | | |
|---|---|---|---|---|---|---|---|---|---|---|---|---|---|---|
| | Mean (SD) | | Estimate of difference (95% CI) STRIP vs DIPP | P | Mean (SD) | | | Estimate of difference (95% CI) STRIP vs DIPP | P | Estimate of difference (95% CI) Boy cohort vs DIPP | P | Estimate of difference (95% CI) Boy cohort vs STRIP | P | Overall P[a] |
| | DIPP N = 1330 | STRIP N = 341 | | | DIPP N = 1495 | STRIP N = 370 | Boy cohort N = 66 | | | | | | | |
| APHV (years)[b] | 11.7 (0.83) | 11.6 (0.83) | -0.07 (-0.20, 0.05) | 0.254 | 13.7 (0.88) | 13.6 (0.86) | 13.8 (1.21) | -0.11 (-0.25, 0.03) | 0.131 | 0.13 (-0.17, 0.42) | 0.396 | 0.23 (-0.07, 0.54) | 0.136 | 0.185 |
| PHV (cm/year)[c] | 7.8 (0.91) | 8.0 (1.04) | 0.13 (0.02, 0.25) | 0.024 | 9.9 (1.09) | 10.3 (1.29) | 10.0 (1.19) | 0.35 (0.21, 0.49) | <0.001 | 0.13 (-0.16, 0.43) | 0.384 | -0.23 (-0.55, 0.10) | 0.169 | <0.001 |
| Estimated height at 8y (cm)[d] | 127.7 (5.34) | 129.5 (5.70) | 1.72 (0.92, 2.51) | <0.001 | 129.8 (5.40) | 130.5 (5.51) | 131.2 (6.04) | 1.24 (0.33, 2.15) | 0.007 | 2.25 (0.33, 4.17) | 0.022 | 1.01 (-1.02, 3.04) | 0.331 | 0.004 |
| Estimated height at 18y (cm)[d] | 165.6 (5.34) | 167.4 (5.70) | | | 179.6 (5.40) | 181.1 (5.51) | 181.8 (6.04) | | | | | | | |

[a] Obtained with Wald test.

[b] Estimates of differences calculated by using delta method on timing parameter.

[c] Estimates of differences calculated by using delta method on intensity parameter.

[d] Estimates of differences are overall differences based on size parameter.

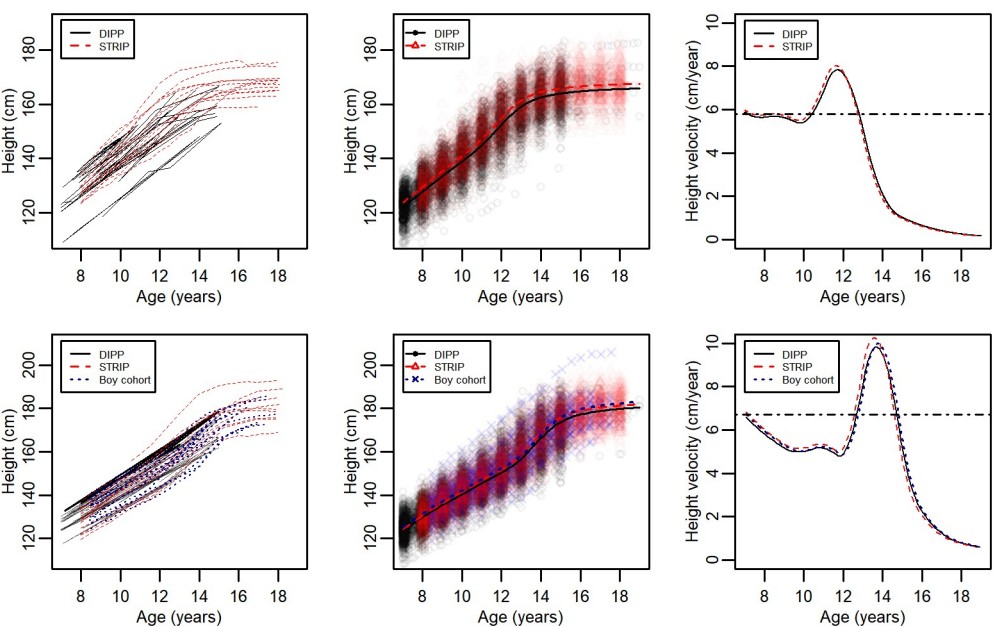

**Fig 2. For girls (above) and boys (below): a) Observed growth for 15 randomly selected children from every cohort, b) SITAR-based cohort-specific average growth curves with height measurements, and c) SITAR-based growth velocity curves.**

overweight status and its interaction with aPHV to the simple model (S1 Table). The modest improvement was likely due to lower aPHVs among overweight girls compared with other girls.

Of the boys, 77 (17.8%) were overweight. The timing of pubertal onset did not differ between boys with and without overweight (−0.0 ([95% CI: −0.3, 0.2] years). Overweight status was still found to be a necessary variable in the boys' extended model. Nevertheless, the estimated regression coefficient of the overweight status (0.2 [95% CI: 0.0, 0.4] years) remained small in the extended model, and hence, it did not improve the overall prediction results dramatically: The agreement increased from 76.0% to 77.0% (S1 Table). Specifically, overweight appeared to be associated with later pubertal onset among boys with larger aPHV values.

## Discussion

We determined the timing of pubertal onset based on growth in terms of individual-specific time intervals representing the period of pubertal onset. In the results, cross-validation showed a good agreement between the observed and predicted timings. The children in Southwest Finland (STRIP and Boy cohort) were taller than the children from Northern Ostrobothnia (DIPP). In addition, STRIP children had higher PHV values compared with DIPP children. Overweight girls had an earlier pubertal onset than normal-weight girls did. In boys, there was no such difference.

Khairullah et al. [16] investigated the relationships between three different height markers, including aPHV, and the pubertal stage at one measurement point, at the age of 13.1–13.8 years. In contrast to the present study, pubertal stage was self-assessed and the association of aPHV with the timing of onset was not considered. Gasser et al. [17] studied the chronology of a variety of pubertal developmental milestones and the associations between them. Associations were studied by determining the proportions of children who reached a specific

milestone at some level of the other milestone. Cole et al. [14] determined the most intense stage of pubertal development based on separate applications of different markers, including height and Tanner G/B. However, these researchers did not study the associations between the markers. Zhu et al. [19] determined the pubertal status (prepubertal, pubertal, and post-pubertal) of individuals at single visits based on the height velocity between visits, growth chart review, and clinical adjudication using predefined threshold velocities [18], and they compared their assessments to the clinical Tanner staging visit by visit. However, they did not use modeling to determine the growth-based pubertal status, and they did not determine the age at pubertal onset or aPHV at all. McCormack et al. [20] presented the association between aPHV, PHV, and age at pubertal onset determined using similar definitions as in the present study. They used the age at the first observation of pubertal onset as a timing of onset and classified ages into three groups, whereas our model used continuous values and considered the interval censoring of the outcome, and provided point and interval estimates of the age at pubertal onset. Their results agree with ours: aPHV was directly and PHV was inversely related to pubertal onset age. Our study provides an alternative model to theirs based on Finnish data.

Since the previous literature indicates an association between increased BMI and age at pubertal onset [23,24,26], we considered it important to include the overweight status of the children in the time-to-pubertal onset model. We found that boys with overweight had a similar age at onset of puberty compared with other boys, but they exhibited an earlier aPHV. This finding is similar to that in a recent study from Denmark [25], where testicular volume $\geq 4$ mL occurred significantly earlier in obese compared with normal-weight boys but Tanner genital stage-based ages at pubertal onset did not differ between those boys. Together, these findings indicate that one possible explanation for the previous contradictory results in boys could be the variation in the chosen pubertal marker. Compared with other girls, those who were overweight exhibited earlier values in terms of both onset of puberty and aPHV. The observations of earlier aPHV values among girls and boys with overweight are consistent with previous literature [39].

The extended time-to-pubertal onset model we used gave only slightly better prediction results than the simple model that included only aPHV and PHV. Thus, using these two growth markers as the only predictors could be appropriate. We think that these parameters need to be determined sufficiently well, and during the study, we found that the SITAR model was well suited for this purpose. We determined the overweight status at the age of the minimum height velocity (based on the SITAR curves) before the growth spurt. In our data, a more straightforward classification by BMI at 2 years before the aPHV would not have resulted in any major changes because the overweight status at these two alternative times agreed nearly perfectly.

We observed a similar difference in height in adolescents between the Oulu (DIPP) and Turku (STRIP, Boy cohort) regions, as was previously presented with adult data [22]. The finding of the higher PHV values in STRIP children compared with DIPP children could be related to the same phenomenon. Some environmental and genetic factors could explain the difference. In addition, type 1 diabetes is known to adversely affect linear growth [40]. Thus, the genetic susceptibility to type 1 diabetes of DIPP participants may be one factor that explains the differences to the DIPP cohort. However, the risk of developing type 1 diabetes with genetic susceptibility is only 1.7%–8.0% depending on the risk genotype [41], and it is unclear whether genetic susceptibility alone affects growth.

The major strengths of the present study were the large, unique dataset, representing a combination of three carefully followed cohorts. Growth was regularly measured during childhood and adolescence in all the cohorts, and pubertal markers were assessed in two of the cohorts. The SITAR model enabled the straightforward and effective production of cohort-specific and

individual-specific growth parameters, and the time-to-pubertal onset model enabled the inclusion of censored pubertal onsets in the model.

The lack of pubertal measurements in the DIPP Study was one limitation of the present research. We had to leave the DIPP Study out of the pubertal analyses, which substantially reduced the amount of data available for the Stage 2 models. Although DIPP children can be a selective sample, the data included children from two other studies, extending the applicability of the results. Cross-validation was used to improve the generalizability of the predictions to external samples.

The estimation of age at pubertal onset is best viewed as an interval estimation: Such estimation provides a prediction interval that can be seen to indicate the range of the most probable ages of pubertal onset. Since real measurements also only provide interval censored data, the interpretation shares similarities with the real measurements (e.g., Tanner stages) taken at fixed times. The determination of pubertal onset timing based on pubertal growth markers is promising method and with replication in other cohorts it could facilitate the determination of the timing in adolescent cohort studies more generally.

## Supporting information

**S1 Fig. Distributions of the aPHV values by the number of the height measurements for girls and boys.**
(DOCX)

**S2 Fig. Associations between growth markers and the age at pubertal onset.** Subject-specific ages at peak height velocity (aPHV), peak height velocities (PHV) and size parameter values plotted against the observed pubertal onset intervals for girls (above) and for boys (below). Passing line is the regression line between the growth marker and the midpoint of the observed pubertal onset interval.
(DOCX)

**S3 Fig. Association between aPHV and the age at pubertal onset for normal weight and overweight girls.** Individual aPHVs plotted against the observed pubertal onset intervals separately for normal weight and overweight girls. Passing line is the regression line between aPHV value and the midpoint of the observed pubertal onset interval.
(DOCX)

**S1 Table. Parameters of the time-to-pubertal onset models, with prediction results.** Parameter estimates of the time-to-pubertal onset models with standard errors in brackets, and agreement between the observed and predicted pubertal onset for simple and extended model.
(DOCX)

**S1 File. Methodological supplementary.**
(DOCX)

## Author Contributions

**Conceptualization:** Harri Niinikoski, Helena E. Virtanen, Jorma Ilonen, Jorma Toppari, Suvi M. Virtanen, Riitta Veijola, Jaakko Peltonen, Jaakko Nevalainen.

**Data curation:** Essi Syrjälä, Helena E. Virtanen, Wiwat Rodprasert, Jorma Toppari, Suvi M. Virtanen, Riitta Veijola.

**Formal analysis:** Essi Syrjälä.

**Funding acquisition:** Harri Niinikoski, Helena E. Virtanen, Mikael Knip, Katja Pahkala, Jorma Toppari, Suvi M. Virtanen, Riitta Veijola.

**Investigation:** Helena E. Virtanen, Mikael Knip, Nina Hutri-Kähönen, Olli T. Raitakari, Jorma Toppari, Suvi M. Virtanen, Riitta Veijola.

**Methodology:** Essi Syrjälä, Harri Niinikoski, Wiwat Rodprasert, Suvi M. Virtanen, Riitta Veijola, Jaakko Peltonen, Jaakko Nevalainen.

**Resources:** Harri Niinikoski, Jorma Ilonen, Mikael Knip, Nina Hutri-Kähönen, Katja Pahkala, Jorma Toppari, Suvi M. Virtanen, Riitta Veijola.

**Supervision:** Harri Niinikoski, Jorma Ilonen, Jorma Toppari, Suvi M. Virtanen, Riitta Veijola, Jaakko Peltonen, Jaakko Nevalainen.

**Validation:** Essi Syrjälä.

**Visualization:** Essi Syrjälä, Jaakko Peltonen, Jaakko Nevalainen.

**Writing – original draft:** Essi Syrjälä.

**Writing – review & editing:** Harri Niinikoski, Helena E. Virtanen, Jorma Ilonen, Mikael Knip, Nina Hutri-Kähönen, Katja Pahkala, Olli T. Raitakari, Wiwat Rodprasert, Jorma Toppari, Suvi M. Virtanen, Riitta Veijola, Jaakko Peltonen, Jaakko Nevalainen.

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
