## [Decision Letter · Decision Letter 0]

13 Sep 2021

PONE-D-21-15712Determining the timing of pubertal onset via a multicohort analysis of growth

PLOS ONE

Dear Dr. Essi Syrjälä,

Thank you for submitting your manuscript to PLOS ONE. After careful consideration, we feel that it has merit but does not fully meet PLOS ONE’s publication criteria as it currently stands. Therefore, we invite you to submit a revised version of the manuscript that addresses the points raised during the review process.

This is professionally written article, but it needs minor revision following review by subject's reviewers. 

The review comments can be found at the end of this email, together with any comments from the Editorial Office regarding formatting changes or additional information required to meet the journal’s policies at this time.

Please note that your revision may be subject to further review and that this initial decision does not guarantee acceptance.

I am sorry that we cannot be more positive on this occasion but hope that you appreciate the reasons for this decision.

We look forward to receiving your revised manuscript.

Kind regards,

Prof Sajid Soofi

Academic Editor

PLOS ONE

Additional Editor Comments (if provided):

Please review the comments of reviewers and revise the paper.

Reviewers' comments:

Reviewer's Responses to Questions

**Comments to the Author**

1. Is the manuscript technically sound, and do the data support the conclusions?

Reviewer #1: Yes

Reviewer #2: Yes

2. Has the statistical analysis been performed appropriately and rigorously? 

Reviewer #1: Yes

Reviewer #2: Yes

3. Have the authors made all data underlying the findings in their manuscript fully available?

Reviewer #1: Yes

Reviewer #2: No

4. Is the manuscript presented in an intelligible fashion and written in standard English?

Reviewer #1: Yes

Reviewer #2: Yes

5. Review Comments to the Author

Reviewer #1: The authors present SITAR analysis of growth relative to pubertal analysis using three Finnish cohort studies—the Type 1 Diabetes Prediction and Prevention (DIPP) Study, the Special Turku Coronary Risk Factor Intervention Project (STRIP), and the Boy cohort. The authors conclude that their novel modeling approach provides an opportunity to evaluate Tanner breast/genital stage–based pubertal onset timing in cohort studies including longitudinal data on growth but lacking pubertal follow-up.

The study represents an important contribution to puberty research but the authors may have overstated the importance of their findings on line 458 and 159 as 'immediate clinical relevance and applicability'. Replication of these results is other dataset is warranted before such a generalization can be made. Perhaps they could suggest that this method is promising and with replication in other cohorts ...

Reviewer #2: Good manuscript takes up available data and utilizes models to draw conclusions

well written but if the models used like SITAR can be explained a bit it would be good for the average reader

Also the data has not been released due to privacy restrictions can that be explained more

6. PLOS authors have the option to publish the peer review history of their article (what does this mean?). If published, this will include your full peer review and any attached files.

Reviewer #1: No

Reviewer #2: **Yes: **Khadija Nuzhat Humayun

---

## [Author Response · Author response to Decision Letter 0]

24 Sep 2021

Response to Reviewers

Reviewer #1: The authors present SITAR analysis of growth relative to pubertal analysis using three Finnish cohort studies—the Type 1 Diabetes Prediction and Prevention (DIPP) Study, the Special Turku Coronary Risk Factor Intervention Project (STRIP), and the Boy cohort. The authors conclude that their novel modeling approach provides an opportunity to evaluate Tanner breast/genital stage–based pubertal onset timing in cohort studies including longitudinal data on growth but lacking pubertal follow-up.

The study represents an important contribution to puberty research but the authors may have overstated the importance of their findings on line 458 and 459 as 'immediate clinical relevance and applicability'. Replication of these results is other dataset is warranted before such a generalization can be made. Perhaps they could suggest that this method is promising and with replication in other cohorts ...

Thank you for the essential remark. We changed the sentence on lines 477-479 (in a manuscript with tracked changes) as “The determination of pubertal onset timing based on pubertal growth markers is promising method to facilitate the determination of the timing in adolescent cohort studies, and with replication in other cohorts this may have immediate clinical relevance and applicability.”

Reviewer #2: Good manuscript takes up available data and utilizes models to draw conclusions well written but if the models used like SITAR can be explained a bit it would be good for the average reader. Also the data has not been released due to privacy restrictions can that be explained more.

Thank you for the relevant comment. We have now clarified the SITAR modeling in the “First stage: Super-Imposition by Translation And Rotation (SITAR) growth curve model” section. We also added a reference to the Supporting S1 File in the end of the SITAR model section and in the end of the “Second stage: Time-to-pubertal onset model” section. S1 File includes the full technical details of the modeling.

We also revised our Data Availability statement as “The dataset supporting the conclusions of this article were obtained from the DIPP, STRIP and Boy cohort studies. The dataset comprises health related participant data and their use is therefore restricted under the regulations on professional secrecy (Act on the Openness of Government Activities, 612/1999) and on sensitive personal data (Personal Data Act, 523/1999, implementing the EU data protection directive 95/46/EC). Due to these legal restrictions, the data from this study cannot be stored in public repositories or otherwise made publicly available. However, data access may be permitted on a case by case basis upon request only. Data sharing outside the research groups is done in collaboration with DIPP, STRIP and Boy cohort groups and requires a group-specific data transfer agreements (DTA). Investigators can submit an expression of interest to the chairman of the DIPP steering committee (Prof Riitta Veijola, University of Oulu, Oulu, Finland. Email: riitta.veijola@oulu.fi), to the chairman of the STRIP steering group (Prof Olli Raitakari, University of Turku, Turku, Finland. Email: olli.raitakari@utu.fi) and to the chairman of the Boy cohort steering group (Prof Jorma Toppari, University of Turku, Turku, Finland. Email: jorma.toppari@utu.fi).”.

---

## [Decision Letter · Decision Letter 1]

15 Oct 2021

PONE-D-21-15712R1Determining the timing of pubertal onset via a multicohort analysis of growthPLOS ONE

Dear Dr. Essi Syrjälä

Thank you for submitting your manuscript to PLOS ONE. After careful consideration, we feel that it has merit but does not fully meet PLOS ONE’s publication criteria as it currently stands. Therefore, we invite you to submit a revised version of the manuscript that addresses the points raised during the review process.

Please address minor comments from one of the reviewers

We look forward to receiving your revised manuscript.

Kind regards,

Prof Sajid Bashir Soofi

Academic Editor

PLOS ONE

Journal Requirements:

Additional Editor Comments:

Please address the blow comments by one of the reviewers

"The authors present SITAR analysis of growth relative to pubertal analysis using three Finnish cohort studies—the Type 1 Diabetes Prediction and Prevention (DIPP) Study, the Special Turku Coronary Risk Factor Intervention Project (STRIP), and the Boy cohort. The authors conclude that their novel modeling approach provides an opportunity to evaluate Tanner breast/genital stage–based pubertal onset timing in cohort studies including longitudinal data on growth but lacking pubertal follow-up.

The study represents an important contribution to puberty research, but the authors may have overstated the importance of their findings on line 458 and 159 as 'immediate clinical relevance and applicability'. Replication of these results is other dataset is warranted before such a generalization can be made. Perhaps they could suggest that this method is promising and with replication in other cohorts"

Reviewers' comments:

Reviewer's Responses to Questions

**Comments to the Author**

1. If the authors have adequately addressed your comments raised in a previous round of review and you feel that this manuscript is now acceptable for publication, you may indicate that here to bypass the “Comments to the Author” section, enter your conflict of interest statement in the “Confidential to Editor” section, and submit your "Accept" recommendation.

Reviewer #1: All comments have been addressed

Reviewer #2: All comments have been addressed

2. Is the manuscript technically sound, and do the data support the conclusions?

Reviewer #1: Yes

Reviewer #2: (No Response)

3. Has the statistical analysis been performed appropriately and rigorously? 

Reviewer #1: Yes

Reviewer #2: (No Response)

4. Have the authors made all data underlying the findings in their manuscript fully available?

Reviewer #1: Yes

Reviewer #2: (No Response)

5. Is the manuscript presented in an intelligible fashion and written in standard English?

Reviewer #1: Yes

Reviewer #2: (No Response)

6. Review Comments to the Author

Reviewer #1: The additional revisions made by the authors have further improved the paper, and I believe that it is acceptable for publication.

Reviewer #2: (No Response)

7. PLOS authors have the option to publish the peer review history of their article (what does this mean?). If published, this will include your full peer review and any attached files.

Reviewer #1: No

Reviewer #2: **Yes: **Khadija Nuzhat Humayun

---

## [Author Response · Author response to Decision Letter 1]

18 Oct 2021

Reviewer comment: The authors present SITAR analysis of growth relative to pubertal analysis using three Finnish cohort studies—the Type 1 Diabetes Prediction and Prevention (DIPP) Study, the Special Turku Coronary Risk Factor Intervention Project (STRIP), and the Boy cohort. The authors conclude that their novel modeling approach provides an opportunity to evaluate Tanner breast/genital stage–based pubertal onset timing in cohort studies including longitudinal data on growth but lacking pubertal follow-up.

The study represents an important contribution to puberty research but the authors may have overstated the importance of their findings on line 458 and 459 as 'immediate clinical relevance and applicability'. Replication of these results is other dataset is warranted before such a generalization can be made. Perhaps they could suggest that this method is promising and with replication in other cohorts ...

Answer: Thank you for bringing this up again. We have now re-changed the sentence on lines 477-480 (in a manuscript with tracked changes) as “The determination of pubertal onset timing based on pubertal growth markers is promising method and with replication in other cohorts it could facilitate the determination of the timing in adolescent cohort studies more generally.”

---

## [Editor Report · Decision Letter 2]

4 Nov 2021

Determining the timing of pubertal onset via a multicohort analysis of growth

PONE-D-21-15712R2

Dear Dr. Essi Syriala

We’re pleased to inform you that your manuscript has been judged scientifically suitable for publication and will be formally accepted for publication once it meets all outstanding technical requirements.

Within one week, you’ll receive an e-mail detailing the required amendments. When these have been addressed, you’ll receive a formal acceptance letter, and your manuscript will be scheduled for publication.

An invoice for payment will follow shortly after the formal acceptance. To ensure an efficient process, please log into Editorial Manager at http://www.editorialmanager.com/pone/, click the 'Update My Information' link at the top of the page, and double check that your user information is up to date. If you have any billing related questions, please contact our Author Billing department directly at authorbilling@plos.org.

Kind regards,

Prof Sajid Soofi

Academic Editor

PLOS ONE
---

## [Editor Report · Acceptance letter]

9 Nov 2021

PONE-D-21-15712R2 

Determining the timing of pubertal onset via a multicohort analysis of growth 

Dear Dr. Syrjälä:

I'm pleased to inform you that your manuscript has been deemed suitable for publication in PLOS ONE. Congratulations! Your manuscript is now with our production department. 

Kind regards, 

on behalf of

Professor Sajid Bashir Soofi 

Academic Editor

PLOS ONE